# MergeMix: Optimizing Mid-Training Data Mixtures via Learnable Model Merging

Jiapeng Wang [1 2 3 *]   Changxin Tian [2 *]   Kunlong Chen [2]   Ziqi Liu [2]   Jiaxin Mao [1 3]
Wayne Xin Zhao [1 3 ✉]   Zhiqiang Zhang [2]   Jun Zhou [2]

## Abstract

Optimizing data mixtures is essential for unlocking the full potential of large language models (LLMs), yet identifying the optimal composition remains computationally prohibitive due to reliance on heuristic trials or expensive proxy training. To address this, we introduce **MergeMix**, a novel approach that efficiently determines optimal data mixing ratios by repurposing model merging weights as a high-fidelity, low-cost performance proxy. By training domain-specific experts on minimal tokens and optimizing their merging weights against downstream benchmarks, MergeMix effectively optimizes the performance of data mixtures without incurring the cost of full-scale training. Extensive experiments on models with 8B and 16B parameters validate that MergeMix achieves performance comparable to or surpassing exhaustive manual tuning while drastically reducing search costs. Furthermore, MergeMix exhibits high rank consistency (Spearman $\rho > 0.9$) and strong cross-scale transferability, offering a scalable, automated solution for data mixture optimization.

## 1. Introduction

Data is the fundamental fuel for large language models (LLMs), with its strategic curation playing a critical role throughout the training lifecycle (Zhao et al., 2023; Albalak et al., 2024; Luo et al., 2025). During pre-training, broad and unlabeled corpora establish foundational linguistic and world knowledge. In mid-training, carefully curated

*Equal contribution   ✉ Corresponding author.
[1] Gaoling School of Artificial Intelligence, Renmin University of China, Beijing, China [2] Ant Group, Hangzhou, China [3] Beijing Key Laboratory of Research on Large Models and Intelligent Governance, Beijing, China. Correspondence to: Wayne Xin Zhao <batmanfly@gmail.com>.

*Proceedings of the $43^{rd}$ International Conference on Machine Learning*, Seoul, South Korea. PMLR 306, 2026. Copyright 2026 by the author(s).

datasets are introduced to enhance specific capabilities, such as reasoning or coding. Finally, post-training stages employ instruction and preference data to align model outputs with human intentions and safety standards. At each stage, the composition of the data mixture directly shapes the model's capabilities, safety, and overall performance (Yang et al., 2025; Hu et al., 2024; Basant et al., 2025; Team et al., 2025a; Olmo et al., 2025).

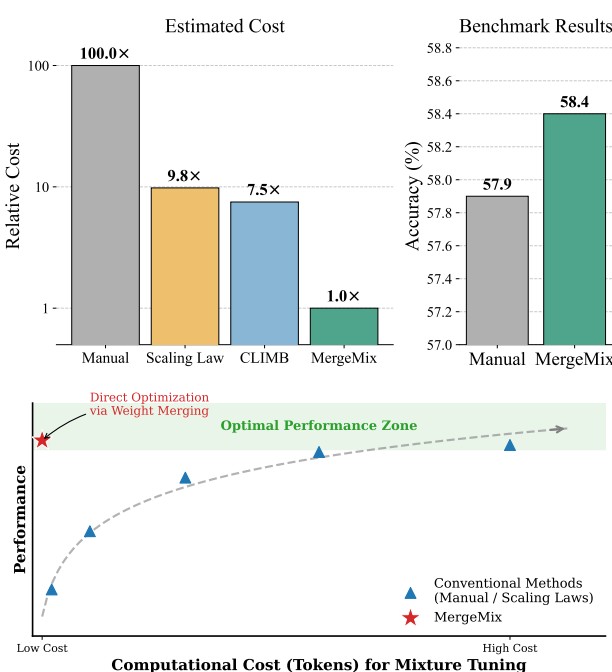

*Figure 1.* Cost-performance efficiency analysis. **(Top)** Comparison of estimated computational costs (log scale) and downstream benchmark accuracy across different data mixing strategies. Details on cost estimation are provided in Appendix C. **(Bottom)** Conceptual illustration of the search dynamics. Conventional methods require increasing computational investment to asymptotically approach the optimal performance zone through iterative trials or fitting. In contrast, MergeMix leverages weight-space merging as an proxy, identifying the optimal mixtures with minimal cost.

Current industrial practices on data mix heavily rely on heuristic trials guided by human intuition, requiring compu-

tationally prohibitive full-scale training runs to identify the best-performing candidate. Although recent studies aim to automate this process, they suffer from some major limitations (Xie et al., 2023; Shukor et al., 2025; Liu et al., 2025; Diao et al., 2025). First, they still incur substantial computational costs, often necessitating dozens to hundreds of proxy training runs to fit scaling laws, train mix regressors, or perform iterative tuning (Shukor et al., 2025; Ye et al., 2025; Liu et al., 2025; Diao et al., 2025). Second, these methods are primarily optimized for language modeling loss (e.g., perplexity) (Xie et al., 2023; Belenki et al., 2025). However, reductions in perplexity do not reliably translate to improved performance on complex downstream tasks (Lourie et al., 2025). This objective misalignment can hinder practical deployment, especially for the suboptimal mixtures used during mid- and post-training. Furthermore, these automatic methods have primarily been evaluated at limited scale and under controlled experimental conditions. This constrains our understanding of their generalization and practical utility, highlighting the need for rigorous, large-scale validation.

To address these limitations, we introduce **MergeMix**, a novel approach that reframes the data mixture optimization problem as a task of *model merging*. We focus particularly on optimizing data mixtures during *mid-training*—the stage where specific model capabilities are refined using carefully curated datasets. Our core insight is that, at this stage, linearly interpolating the weights of domain-specific expert models serves as a high-fidelity proxy for the outcomes of actual data mixing. Rather than running expensive training trials for every candidate mixture ratio, MergeMix requires training only a small set of experts on minimal data. By optimizing the merging weights of these experts against downstream benchmarks, we effectively convert the high cost of model training into the negligible cost of model merging and inference. We validate MergeMix using industrial-scale datasets, conducting extensive mid-training experiments on models with 8B and 16B parameters. Our experiments show that MergeMix identifies data configurations that match or exceed the performance of exhaustive manual tuning, while reducing the search cost by over $100\times$.

To summarize, our contributions are as follows:

- We propose **MergeMix**, a novel approach that leverages model merging as a computationally efficient and high-fidelity proxy for evaluating data mixtures. This allows us to optimize data configurations directly against downstream tasks at a drastically reduced cost.
- We provide a theoretical analysis on why weight interpolation between domain-expert models effectively approximates the outcome of direct data-mixture tuning. This is grounded by the identical first-order optimization dynamics between weight interpolation and mixed-data training under shared initialization.

- We conduct extensive experiments to validate MergeMix in industrial-scale mid-training scenarios. Our results show that MergeMix achieves performance comparable to or better than extensive manual tuning, while reducing the computational cost of the search process by $100\times$.

## 2. Related Work

### 2.1. Model Merging

Model merging (Izmailov et al., 2018; Wortsman et al., 2022) has recently gained attention as a parameter-space alternative to ensembling or retraining for improving performance and reusing existing models efficiently. It has been shown that averaging weights of multiple models from the same base often yields a single model with higher accuracy and robustness than the best individual model without increasing inference cost (Wortsman et al., 2022), and recent advancements have moved beyond simple averaging to sophisticated weighted combinations aimed at maximizing cross-domain performance (Maiti et al., 2025; Khalifa et al., 2024). Building on this foundation, a growing body of work has explored diverse merging algorithms to further refine how model parameters are combined (Ilharco et al., 2023; Yu et al., 2024; Yadav et al., 2023; Yang et al., 2024). Ahmadian et al. (2024) explores the effectiveness of objective-driven model merging and mixed-data training. More closely related to our work, several recent studies (Na et al., 2024; Tao et al., 2025) explore model merging as a means to ablate data inclusion, but do not investigate the impact of data mixing ratios.

### 2.2. Data Mixture for LLM Pre-training

The strategic allocation of training data plays a pivotal role in shaping the performance characteristics of LLMs. Recent progress on pre-training data mixture optimization has shifted from heuristic sampling toward predictive, automated, and scalable strategies. A notable strand uses scaling laws to model and predict performance across mixture proportions and scales (Ye et al., 2025; Ge et al., 2024; Gu et al., 2024b; Shukor et al., 2025). Another theme focuses on proxy models and automated search for mixture optimization, including DoReMi (Xie et al., 2023) that uses small proxy models to reweight domain proportions, RegMix (Liu et al., 2025) that treats mixture search as a regression prediction task over many small runs, ADMIRE-BayesOpt (Chen et al., 2025) which formulates mixture selection as a Bayesian optimization problem for accelerated search across model scales, CLIMB (Diao et al., 2025), a clustering-based iterative bootstrapping framework that embeds data, evaluates candidate mixtures with proxies, and refines weights, and MixMin (Thudi et al., 2025), which formulates the mixture search as a convex minimization problem to efficiently match target data distributions. Most

closely related to our work is Belenki et al. (2025), which ensembles multiple domain-specialized experts by combining their logits to approximate the loss on mixed-domain data. However, this approach primarily optimizes for validation loss and incurs an inference cost that scales linearly with the number of experts. Moreover, its effectiveness has not been validated in real-world training scenarios or across diverse downstream benchmarks.

## 3. The MergeMix Framework

The proposed MergeMix framework operates on the premise that in the mid-training phase, the parameter space geometry is sufficiently regular (Frankle et al., 2020) that *weight interpolation* can serve as a computationally efficient proxy for *data interpolation*. The MergeMix pipeline consists of three stages: First, we train expert models to capture the optimization trajectory of each specific domain. Next, we employ a regression model to approximate the relationship between weight mixing ratios and downstream capabilities, allowing us to efficiently explore the performance landscape without expensive training trials. Finally, we identify the data mixture that maximizes the target utility and apply this derived mixture to the large-scale training run.

### 3.1. Training Domain-Specific Experts

Given a pretrained base model $\Theta_{\text{base}}$ and $K$ data domains $\{\mathcal{D}_1, \ldots, \mathcal{D}_K\}$, we train $K$ independent expert models. Our training procedure incorporates the following key configurations: (1) Shared initialization: All experts are initialized from the same base model $\Theta_{\text{base}}$; (2) Constant learning rate: A fixed learning rate $\eta$ is applied throughout training without decay. This prevents the vanishing update step size typical of LR decay, ensuring that the expert models continuously traverse the loss landscape along their respective domain-specific gradient directions; (3) Restricted training horizon: Training is limited to a short horizon (e.g., fewer than 5B tokens). This keeps each expert within a local neighborhood around the initialization, preserving the geometric alignment required for effective weight merging and preventing divergence into separate loss basins (Frankle et al., 2020).

### 3.2. Efficient Mixture Search via Model Merging

We reframe the costly mixture-tuning problem into a model merging process. This enables the highly efficient instantiation of candidate mixture models. Instead of relying on inaccurate loss-based predictors, we directly evaluate and optimize the merged models on downstream benchmarks.

**Parameter-Space Merging.** Given a mixing configuration vector $\boldsymbol{\alpha} \in \Delta^{K-1}$ (where $\sum \alpha_k = 1$), we instantiate a candidate model $\Theta_{\text{merge}}(\boldsymbol{\alpha})$ via linear interpolation (Iz-

mailov et al., 2018; Wortsman et al., 2022):

$$\Theta_{\text{merge}}(\boldsymbol{\alpha}) = \Theta_{\text{base}} + \sum_{k=1}^{K} \alpha_k (\Theta_k - \Theta_{\text{base}}).$$

Since this operation involves only element-wise addition, it incurs negligible computational cost compared to training.

**Performance Surface Mapping.** Directly evaluating every possible combination on the simplex is prohibitively expensive given the cost of full-benchmark evaluation. Instead, we adopt a learnable prediction strategy: We sample $N$ seed configurations $\{\boldsymbol{\alpha}^{(i)}\}_{i=1}^{N}$ using a coarse grid search combined with heuristic priors ($N = 40$ in our experiments). For each configuration, we construct the merged model, evaluate it across $M$ capability domains (e.g., mathematics, coding, reasoning, knowledge), and record the resulting capability scores $\mathbf{y}^{(i)} \in \mathbb{R}^M$. We then train a set of Light-GBM regressors (Ke et al., 2017) $\hat{f}_m : \boldsymbol{\alpha} \to y_m$ (one per capability) to approximate the performance landscape:

$$\hat{\mathbf{y}}(\boldsymbol{\alpha}) = [\hat{f}_1(\boldsymbol{\alpha}), \hat{f}_2(\boldsymbol{\alpha}), \ldots, \hat{f}_M(\boldsymbol{\alpha})].$$

Using this learned prediction model $\hat{f}$, we can perform a fine-grained search over the simplex to identify the candidate $\boldsymbol{\alpha}^*$ that maximizes our target utility function. The predicted optimum is finally verified via actual model merging and evaluation to ensure reliability.

**Utility-Driven Mixture Selection.** With the performance surface $\hat{f}$ mapped, selecting the optimal mixture becomes a flexible optimization task governed by a user-defined utility function $U(\cdot)$. This framework allows practitioners to define objectives tailored to specific needs, ranging from a balanced generalist model to specialized targeting of distinct capabilities (e.g., coding or reasoning). Consequently, MergeMix enables the efficient exploration of capability trade-offs without the need for training trials. We formalize this selection process as:

$$\boldsymbol{\alpha}^* = \arg \max_{\boldsymbol{\alpha} \in \Delta^{K-1}} U(\hat{f}_1(\boldsymbol{\alpha}), \ldots, \hat{f}_M(\boldsymbol{\alpha})).$$

The derived weight ratios $\boldsymbol{\alpha}^*$ are then directly adopted as the data mixing ratio $\boldsymbol{\lambda}^*$. In our primary experiments, we instantiate $U$ as a generalist objective, calculated as the macro-average of normalized benchmark scores, to ensure robust and balanced improvements across all domains. We also present results focused on optimizing specific capabilities in Figure 5.

### 3.3. Theoretical Analysis: Why Weight Mixing Proxies Data Mixing

In this section, we provide a theoretical analysis to justify substituting computationally expensive mixed-data training

with efficient weight interpolation. Our core argument is that in the local mid-training regime, these two processes share identical first-order dynamics, with discrepancies confined to second-order under the specific training protocols defined in Section 3.1.

**Trajectory Decomposition via Taylor Expansion.** Let $\Theta_0$ be the pretrained initialization. We analyze an idealized SGD-like dynamics under the assumption that the change in loss can be approximated by decomposing it into first-order gradient contributions and second-order curvature interactions (Wang et al., 2025b; Pruthi et al., 2020).

First, consider the data mixing trajectory with ratios $\boldsymbol{\lambda}$. The parameter update accumulates gradients from the mixed loss $\mathcal{L}_{\text{mix}} = \sum \lambda_k \mathcal{L}_k$. The final parameters $\Theta^{\text{mix}}$ can be approximated as (see derivation in Appendix B):

$$\Theta^{\text{mix}} \approx \Theta_0 - \eta T \sum_k \lambda_k g_k + \frac{(\eta T)^2}{2} \sum_{k,j} \lambda_k \lambda_j H_k g_j. \quad (1)$$

where $g_k = \nabla \mathcal{L}_k(\Theta_0)$ and $H_k = \nabla^2 \mathcal{L}_k(\Theta_0)$. The second term represents the interaction between domains, where the gradient update of domain $j$ is distorted by the curvature of domain $k$ (Wang et al., 2025b; Yu et al., 2020).

Next, consider the model merging trajectory. We train $K$ experts independently, where the $k$-th expert's update is $\Delta\Theta_k \approx -\eta T g_k - \frac{(\eta T)^2}{2} H_k g_k$. By merging these experts with weights $\boldsymbol{\alpha}$ set equal to the data ratios $\boldsymbol{\lambda}$, we obtain:

$$\Theta^{\text{merge}} \approx \Theta_0 - \eta T \sum_k \lambda_k g_k + \frac{(\eta T)^2}{2} \sum_k \lambda_k H_k g_k. \quad (2)$$

Comparing Eq. (1) and Eq. (2) reveals that the first-order gradient terms match perfectly. The discrepancy is strictly confined to the second-order terms (see detailed discussion of the error term $\Delta$ in Appendix B).

**Validity of Proxy.** The approximation is motivated by the dominance of first-order training dynamics in the local optimization regime. Crucially, recent theoretical analysis indicates that such higher-order contributions offer limited additional signal for data contribution estimation compared to first-order gradient alignment (Wang et al., 2025b; Pruthi et al., 2020; Yeh et al., 2022). In this work, we therefore focus on first-order effects and treat higher-order terms as secondary, using optimization over the merged weights $\boldsymbol{\alpha}$ as a principled and efficient proxy for navigating the data mixing search space.

### 3.4. Computational Cost Analysis

We further quantify the efficiency of MergeMix by analyzing its computational complexity. Let $C_{\text{train}}$ be the training

cost per token and $D_{\text{trial}}$ be the budget allocated for each exploratory training run. Standard manual tuning requires training $N$ candidate mixtures, incurring a total cost of approximately $\mathcal{C}_{\text{Manual}} \approx N \cdot D_{\text{trial}} \cdot C_{\text{train}}$. Even proxy-based methods (e.g., scaling laws) rely on numerous sub-scale runs, where the accumulated token count remains substantial. In contrast, MergeMix requires only training $K$ domain experts on a minimal subset $D_{\text{expert}} \ll D_{\text{trial}}$. Since model merging is computationally negligible, the dominant cost becomes the inference-based evaluation of merged candidates, denoted as $C_{\text{eval}}$. The total cost is thus $\mathcal{C}_{\text{MergeMix}} \approx K \cdot D_{\text{expert}} \cdot C_{\text{train}} + M \cdot C_{\text{eval}}$, where $M$ denotes the number of merged models used for grid search or proxy model training. Given that $C_{\text{eval}}$ is orders of magnitude lower than training costs, and in our setting, where $K = 4$, $N = 10$, and $D_{\text{expert}} \approx 0.025\, D_{\text{trial}}$ (5B vs. 200B tokens), MergeMix achieves a cost reduction of $100\times$ compared to exhaustive manual baselines, which consume 2T tokens versus MergeMix's 20B tokens.

### 3.5. Hierarchical Data Mix

In industrial mid-training, practitioners typically work with hundreds of distinct datasets, not just a few high-level categories, making direct optimization over a high-dimensional simplex intractable. To navigate this, a hierarchical model merging strategy can be adopted. First, semantically similar datasets can be grouped into clusters (e.g., within the *code* domain, sub-clusters such as code repositories, code-related text, and programming competitions). Then, the MergeMix framework can be recursively applied.

This hierarchical search can be executed in two ways: (1) *bottom-up*, where optimal mixing ratios are first determined within each sub-cluster group to form consolidated domain experts, which are subsequently merged at the global level; or (2) *top-down*, where the global mixing ratios across high-level domains are optimized first, followed by independent search of optimal local ratios within each domain. This divide-and-conquer strategy enables precise, fine-grained control over the data distribution without the combinatorial explosion of the search space. We provide empirical experiment of hierarchical MergeMix in Section 4.6.

## 4. Experiment

### 4.1. Experimental Setup

**Models and Training Settings.** We conduct experiments using two models based on a standard Mixture-of-Experts (MoE) architecture. They have 8B and 16B total parameters, respectively, with each activating 1.4B parameters. We refer to them as *small* and *large* in the following sections, respectively. The detailed model architecture is shown in Appendix A. We focus on mid-training with high-quality

data starting from pretrained checkpoint. We employ the AdamW optimizer (Loshchilov & Hutter, 2019), with hyperparameters set to $\beta_1 = 0.9$, $\beta_2 = 0.95$, and a weight decay of 0.1. Based on preliminary scaling law experiments, we set the peak learning rate and batch size to $3.74 \times 10^{-4}$ and 2048 for the large model, and $4.78 \times 10^{-4}$ and 2048 for the small model, respectively.

**Benchmarks.** To provide a holistic assessment of model capabilities, we consider a diverse suite of downstream tasks for evaluation. Tasks are grouped into several categories, including: (a) general knowledge/reasoning (ARC (Bhakthavatsalam et al., 2021), AGIEval (Zhong et al., 2024), OpenBookQA (Mihaylov et al., 2018), BBH (Suzgun et al., 2023), WorldSense (Hong et al., 2025), PIQA (Bisk et al., 2020), hellaswag (Zellers et al., 2019) and KOR-Bench (Ma et al., 2025)); (b) language understanding (race (Lai et al., 2017), SQuAD 2.0 (Rajpurkar et al., 2018), TriviaQA (Joshi et al., 2017), NQ (Kwiatkowski et al., 2019) and winogrande (Sakaguchi et al., 2021)); (c) professional knowledge (e.g., MMLU (Hendrycks et al., 2021a), CMMLU (Li et al., 2024a), C-Eval (Huang et al., 2023), MMLU-Pro (Wang et al., 2024), GPQA (Rein et al., 2023) and SuperG-PQA (Team et al., 2025b)); (d) math (GSM8K (Cobbe et al., 2021), MATH (Hendrycks et al., 2021b), gaokao (Zhang et al., 2023), GSM-Plus (Li et al., 2024b), mgsm-zh (Shi et al., 2023), CMATH (Wei et al., 2023), MathBench (Liu et al., 2024), minerva_math (Hendrycks et al., 2021b) and college_math (Tang et al., 2024); (e) code (HumanEval (Chen et al., 2021), LiveCodeBench (Jain et al., 2025), MBPP (Tao et al., 2024), HumanEval_plus (Liu et al., 2023), MBPP_plus (Liu et al., 2023), HumanEval_cn (Peng et al., 2024), HumanEval_fim (Bavarian et al., 2022) and CruxEval (Gu et al., 2024a)).

### 4.2. Dual-Capability Merging Study

The core premise of MergeMix is that the performance landscape in model weight space mirrors the performance landscape in data mixture ratio space. To validate the MergeMix framework, we first conduct a controlled dual-capacity experiment to rigorously verify the correlation between model merging weights and data mixing ratios.

Specially, we first examine this hypothesis in two domains known to exhibit strong correlation: math and code, where prior works suggest that training on one data type can impact performance on the other (Ma et al., 2024; Lu et al., 2025). First, we train two domain-specialized expert models, $\Theta_{\text{math}}$ and $\Theta_{\text{code}}$, on their respective datasets for 25B tokens each. We then construct two sequences of models: (1) Weight interpolated models: $\Theta_\alpha = \alpha \cdot \Theta_{\text{math}} + (1 - \alpha) \cdot \Theta_{\text{code}}$, $\alpha \in \{0.1, 0.2, \ldots, 0.9\}$; (2) Data mixture trained models: Trained from scratch on

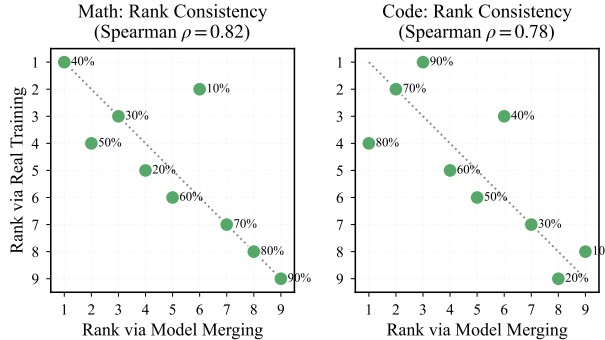

*Figure 2.* Rank consistency between model merging and data mixture training. The high correlation indicates that weight interpolation accurately predicts the relative ranking of data mixtures. We also present the value of $\lambda$ for each configure in percent.

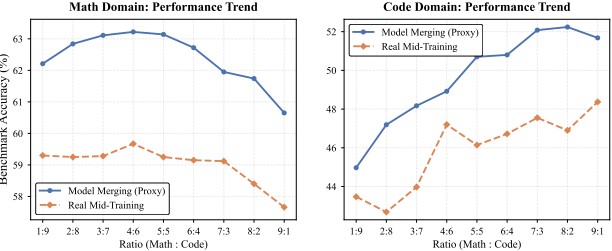

*Figure 3.* Performance trend comparison between model merging and actual mixture tuning on math and code benchmarks.

mixed datasets $\mathcal{D}_\lambda \leftarrow \lambda \cdot \mathcal{D}_{\text{math}} + (1 - \lambda) \cdot \mathcal{D}_{\text{code}}$[1], with $\lambda \in \{0.1, 0.2, \ldots, 0.9\}$, for a total of 25B tokens per model.

We evaluate both configuration sequences on standard math and code benchmarks. As shown in Figure 3, the performance curves of the merged and trained models display highly synchronized trends, with two notable distinctions: (1) there is a consistent offset in performance scale between them, yet their relative trends across mixture ratios remain closely aligned; and (2) the performance curves of the merged models are noticeably smoother. We attribute these differences to the inherent advantages of model merging, which has been shown to mitigate training fluctuations and can even enhance final performance (Wang et al., 2025a; Tian et al., 2025). We argue that the smoothness of model merging curves better reflects the underlying data–performance relationship and may provide a more stable and reliable proxy for ablation studies than full retraining compared to the fluctuations inherent in real training.

Considering the performance offset involved in two performance curves, we further assess the rank consistency.

---

[1]Without sacrificing clarity, we adopt the same interpolation formulation to represent the combined datasets sampled from $\mathcal{D}_{\text{math}}$ and $\mathcal{D}_{\text{code}}$, with sampling ratio $\lambda$ and $1 - \lambda$ respectively.

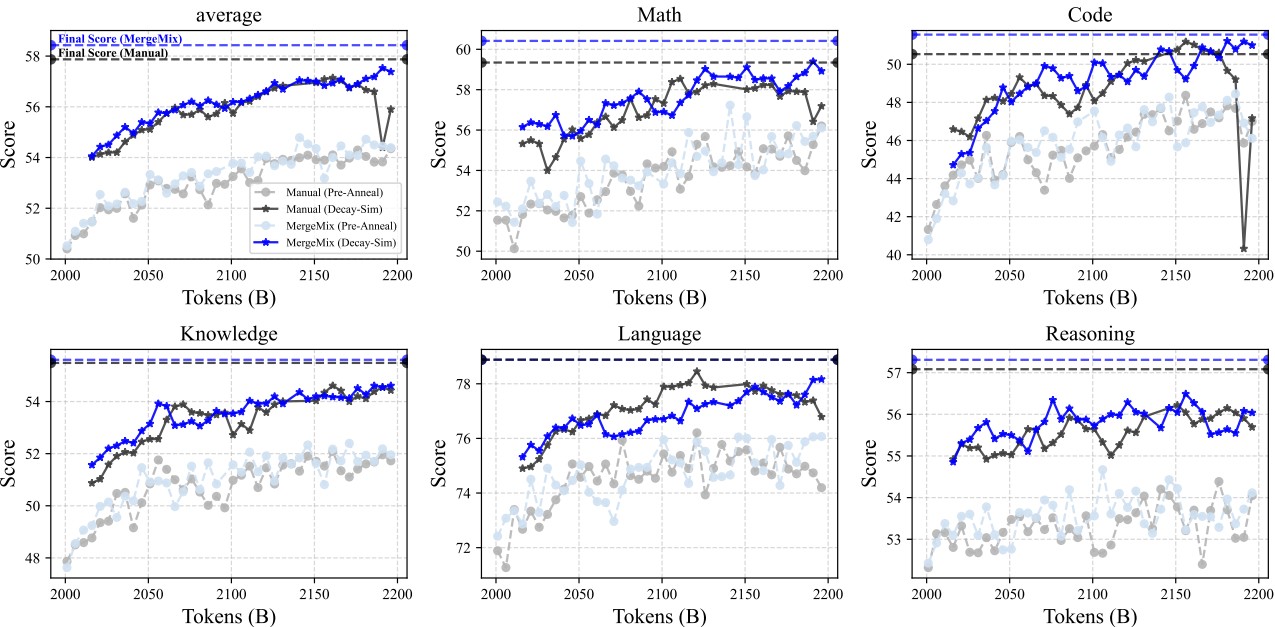

*Figure 4.* Training dynamics comparison across five domains. The light-colored curves (Pre-Anneal) track the performance of models trained with a constant learning rate. The dark-colored curves (Annealed) represent the performance after applying learning rate annealing (simulated by merging the most recent 20B tokens). The horizontal dashed lines denote the final performance by merging the top-16 checkpoints. The model trained with MergeMix-derived ratios consistently matches or outperforms the strong manually tuned baseline.

Figure 2 plots the rank of the trained models against the rank of the merged models across different mixtures. We observed a strong correlation. This implies that if specific merging ratios are optimal in the weight space, the corresponding data mixture is highly likely to be optimal in the training space. It is noteworthy that our method does not aim to predict absolute scores; instead, it predicts the relative ranking of mixtures to identify configurations that are comparatively better. This strong rank consistency justifies using the computationally efficient weight space as a surrogate for the expensive data space.

### 4.3. Large-Scale Mid-Training Validation

**Comparison Setup.** We proceed to validate the MergeMix framework in a large-scale mid-training setting. Following Nemotron's data partitioning strategy (Basant et al., 2025), we categorize the mid-training corpus into four primary domains for mixture ratio tuning: mathematics, code, supervised fine-tuning (SFT), and web/others data. We employ the *small* MoE model, pre-trained on 2T tokens, and allocate a 200B-token mid-training budget. Following the WSM schedule (Tian et al., 2025), we train with a constant learning rate and use model merging to simulate the performance after model annealing. Specifically, we merge the best 16 checkpoints (ranked by benchmark performance) to further boost model performance and use this merged model to represent the final performance. We

compare MergeMix against three baselines: (1) *Uniform*: Uniform sampling across the four categories proportional to their total token counts; (2) *Manual*: A manually tuned ratio derived from extensive ablation studies and heuristic tuning by our data team, which has already been deployed in production for our previous internal flagship model; (3) *Adapted RegMix* (Liu et al., 2025): A regressor-based predictor trained on the full-scale exploration runs collected from the manual baseline to optimize optimal mixtures. More details about baselines are shown in Appendix C.

**Performance and Efficiency.** Table 1 and Figure 4 present the performance comparison across key benchmarks between the model trained with the MergeMix-derived mixture and baselines. Remarkably, MergeMix matches or surpasses the manually tuned baseline across nearly all capability domains, delivering clear gains in math, code, and reasoning, while maintaining near-lossless performance in the remaining domains. Most critically, this result was achieved with a computational cost reduction of over **100×** compared to the exhaustive ablation process required for the baseline. By substituting expensive full-scale trial runs with low-cost model merging inference, MergeMix enables highly efficient identification of optimal data mixtures.

**Predictive Fidelity: Ranking Consistency.** To ensure the reliability of our proxy, we additionally sample 12 distinct

*Table 1.* Performance comparison of different data mixing methods across domains.

| Method | Math | Code | Knowledge | Language | Reasoning | Average |
|---|---|---|---|---|---|---|
| Uniform | 57.1 | 44.6 | 50.0 | 76.0 | 56.7 | 54.2 |
| Adapted RegMix | 59.7 | 51.5 | 55.5 | 78.3 | 57.0 | 58.1 |
| Manual | 59.3 | 50.5 | 55.5 | **78.9** | 57.1 | 57.9 |
| MergeMix | **60.4** | **51.6** | **55.6** | **78.9** | **57.3** | **58.4** |

*Table 2.* Rank consistency (Spearman $\rho$) between MergeMix predictions and actual training outcomes across domains.

| Domain | Overall | Math | Code | Knowledge | Language | Reasoning |
|---|---|---|---|---|---|---|
| Spearman $\rho$ | 0.92 | 0.84 | 0.80 | 0.98 | 0.72 | 0.89 |

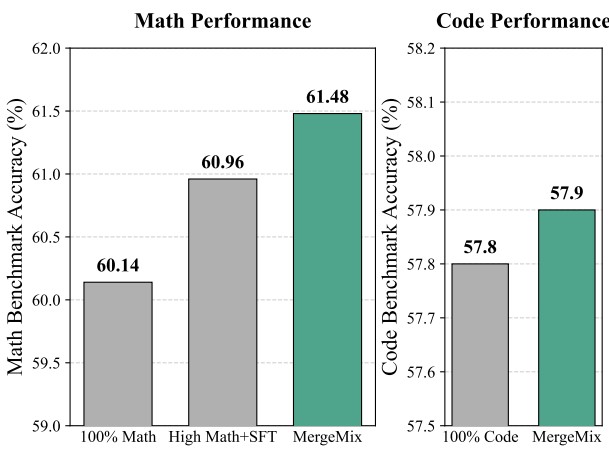

*Figure 5.* Performance comparison between the MergeMix-optimized mixture, uni-domain baselines (100% single-domain data), and an aggressive specialization mixture.

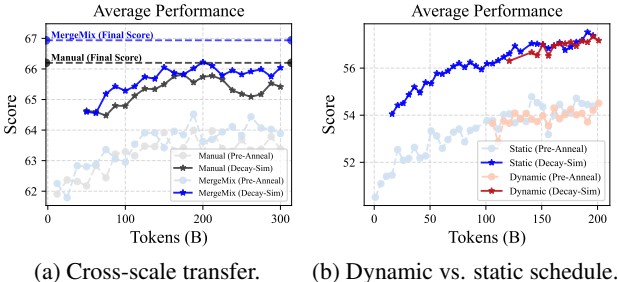

(a) Cross-scale transfer.  (b) Dynamic vs. static schedule.

*Figure 6.* Cross-scale transfer from an 8B proxy to 16B target model (left) and dynamic data mix schedule comparison (right).

mixtures and perform full mid-training runs to verify the predictions. Table 2 reports the Spearman rank correlation ($\rho$) between the predicted rank (via merged weights) and the actual mid-training outcomes. We observe high correlation ($\rho > 0.9$) on overall scores, confirming that the weight-space geometry accurately captures the dynamics of multi-domain data mixing. Notably, the prediction is nearly perfect for knowledge-related metrics ($\rho = 0.98$).

**Landscape Analysis: Dissecting Capability Synergy.** Finally, MergeMix enables a holistic view of the correlating dynamics between data sources. Figure 8 visualizes the performance heatmaps projected onto the weight simplex, revealing distinct topological patterns. For example, code exhibits high domain orthogonality with performance concentrated near its pure-domain composition, while the optimal math region centers around a mixed distribution, suggesting that reasoning is a composite capability requiring synergy between domain knowledge (math), instruction adherence (SFT), and broad linguistic understanding (web). To validate the superiority of this identified mixture, we focus on single domain and compare the MergeMix-derived configuration against uni-domain baselines (e.g., 100% math, 100% code) and an aggressive specialization mixture (high math/SFT, minimal web). As shown in Figure 5, the MergeMix-predicted mixture consistently outperforms these baselines. This empirically verifies that weight-space exploration accurately locates the "sweet spot" of data composition, harnessing the synergy between domains that heuristic or domain-specialization strategies fail to capture.

### 4.4. Cross-Scale Transfer of Data Mixtures

Industrial computational budgets are often tight, especially for intensive mixture-tuning experiments. A key practical advantage would be the ability to identify an optimal data mixture with a lightweight proxy model and then transfer it effectively to a larger and more expensive target model. To investigate this, we conduct a data mix transfer experiment. Instead of running the MergeMix pipeline on the *large* model (16B), we utilize the optimal mixture derived exclusively from the *small* model (8B). We then apply this ratio directly as the mixture for the mid-training of the 16B model. The performance of the large model trained with the small model derived ratio is reported in Figure 6a. We observe that the mixture remains highly effective at the larger scale, even outperforming the manually tuned baseline. This cross-scale consistency further reduces computational overhead, enabling practitioners to conduct extensive exploration in weight space using economical, small-scale models to

*Table 3.* Performance comparison of fine-grained mixing strategies. Notation $\langle A, B \rangle$ denotes using strategy $A$ for coarse-level domain mixture and strategy $B$ for fine-grained sub-cluster mixture. Remarkably, the fully automated MergeMix approach ($\langle$MergeMix, MergeMix$\rangle$) outperforms the fully manual baseline.

| Mixing Strategy | Math | Code | Knowledge | Language | Reasoning | Average |
|---|---|---|---|---|---|---|
| $\langle$Manual, Manual$\rangle$ | 59.3 | 50.5 | 55.5 | 78.9 | 57.1 | 57.9 |
| $\langle$MergeMix, Uniform$\rangle$ | 58.7 | 51.1 | 55.2 | 77.9 | 56.5 | 57.6 |
| $\langle$MergeMix, Manual$\rangle$ | 60.4 | 51.6 | 55.6 | 78.9 | 57.3 | **58.4** |
| $\langle$MergeMix, MergeMix$\rangle$ | 59.8 | 51.3 | 55.3 | 79.1 | 57.2 | 58.2 |

identify effective data mixtures, which can then be applied to larger-scale models.

## 4.5. Dynamic vs. Static Mixtures

Given that model capabilities evolve continuously during training, we investigate whether periodically recalibrating the data mixture yields better performance than maintaining a static mix fixed at initialization. Specially, we conduct a two-stage mixture adjustment to investigate dynamic recalibration. We compare two strategies: (1) a static schedule, where the data mixture optimized at initialization is used throughout the training; and (2) a dynamic schedule, where the mixture is re-optimized at 50% training progress using MergeMix on the intermediate checkpoint, and updated for the second half. As shown in Figure 6b, we find that the re-optimized mixture at the midpoint remains remarkably similar to the initial configuration, resulting in negligible performance differences between the two schedules. It suggests that the optimal data mixture is a stable, intrinsic property of the mid-training task. This stability implies that the optimization landscape does not shift significantly enough to warrant dynamic scheduling. Consequently, a single, static MergeMix search at initialization seems to be sufficient for mid-training, validating our method as a robust and efficient solution without the need for costly iterative re-calibration.

## 4.6. Hierarchical MergeMix

*Table 4.* Fine-grained categorization by primary domain.

| Coarse-level Domain | Fine-Grained Sub-clusters |
|---|---|
| **Mathematics** | Web Math, Math QA, Synthetic Study Notes, Formal Math |
| **Code** | Code Repos, Code NLP, Code Contest |
| **SFT** | Exam, General SFT, Long Chain-of-Thought |
| **Web / Others** | English Web, Chinese Web, Books, Wikipedia, Academic Papers |

In the previous experiments, we operate within a simplified setting of four coarse domains, where intra-domain mixtures are manually set. In this section, we advance toward

fine-grained mixture optimization to further reduce manual intervention and facilitate practical deployment. A straightforward approach would be to treat fine-grained groups of datasets as direct input to MergeMix. However, this would drastically expand the dimensionality of the search space, making the optimization process both less efficient and less effective. To address this, we extend MergeMix into a hierarchical divide-and-conquer framework. As discussed in Section 3.5, this hierarchy can be traversed in two directions. In this experiment, we adopt a top-down strategy: we first optimize the coarse-level mixture based on the optimal values derived in Section 4.3, and subsequently refine the fine-grained mixture within each domain.

We decompose the four coarse domains into 16 distinct sub-clusters (taxonomy detailed in Table 4). The performance comparison is presented in Table 3. Remarkably, the fully automated strategy (denoted by $\langle$*MergeMix, MergeMix*$\rangle$) outperforms the fully manual baseline (i.e., $\langle$*Manual, Manual*$\rangle$). Our method successfully identifies high-value fine-grained sub-clusters, such as assigning higher weights to Math QA and Exam datasets without any human prior. With the coarse-level distribution fixed, MergeMix optimizes fine-grained weights to outperform the uniform baseline (i.e., $\langle$*MergeMix, Uniform*$\rangle$) by 0.6%, achieving performance comparable to exhaustive manual fine-grained tuning (i.e., $\langle$*MergeMix, Manual*$\rangle$).

## 5. Conclusion

In this work, we introduce MergeMix, a resource-efficient framework for optimizing mid-training data mixtures for LLMs. By theoretically and empirically establishing that linear weight interpolation serves as a high-fidelity proxy for data gradient accumulation, we transform the computationally prohibitive problem of data mixture tuning into a low-cost model merging optimization task. Our extensive validation shows that MergeMix identifies data configurations that match or exceed the performance of exhaustive manual tuning and current automated methods, while reducing search costs by orders of magnitude. The framework demonstrates strong rank consistency and cross-scale transferability. MergeMix provides a scalable, automated path-

way to enhancing model capabilities, shifting the paradigm of data composition from heuristic guesswork toward precise, objective-driven engineering.

## Acknowledgements

This work was partially supported by the National Natural Science Foundation of China No. 92470205 and Beijing Major Science and Technology Project under Contract No. Z251100008425002. This work was also supported by Ant Group Research Fund. Xin Zhao is the corresponding author.

## Impact Statement

This paper presents work whose goal is to advance the field of Machine Learning. There are many potential societal consequences of our work, none which we feel must be specifically highlighted here.

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

## A. Model Architecture

The core architectures of our experimental 8B MoE model are detailed in Table 5. The model is configured with 20 layers and a hidden dimension size of 2048. Except for the first layer, all FFNs layers are replaced with MoE layers. We adopt the GQA attention mechanism (Ainslie et al., 2023) and integrate Rotary Position Embedding (RoPE) (Su et al., 2024), enabling the model to support sequence lengths up to 8K tokens. For parameter initialization, all learnable parameters are randomly initialized using a standard deviation of 0.006. Under this configuration, the model consists of a total of 8.6 billion parameters, of which approximately 1.43 billion are activated for each token during inference.

*Table 5.* Detailed model architecture.

| Parameter | Value |
| --- | --- |
| Number of layers ($n_{layers}$) | 20 |
| Model dimension ($d_{model}$) | 2,048 |
| FFN dimension ($d_{ffn}$) | 5,120 |
| Expert dimension ($d_{expert}$) | 512 |
| Number of attention heads ($n_{heads}$) | 16 |
| Number of KV heads ($n_{kv\_head}$) | 4 |
| Total experts ($E$) | 32 |
| Active experts ($E_a$) | 8 |
| Shared experts ($E_s$) | 1 |
| Total parameters ($N$) | 8.6B |
| Active parameters ($N_a$) | 1.43B |

## B. Analysis of the Weight Mixing Proxy

In this appendix, we provide the derivation supporting the claims made in Section 3.3. We formally characterize the discrepancy between the mixed-data training trajectory and the model merging approximation, showing that the error is confined to second-order interactions.

### B.1. Derivation of the Interaction Error

Let $\Theta_0$ be the initialization. We compare the parameters after $T$ steps (where the local quadratic approximation holds) under two regimes: data mixing and model merging.

**Data Mixing Trajectory.** For a data mixture with ratios $\boldsymbol{\lambda} \in \Delta^{K-1}$, the total loss is $\mathcal{L}_{\text{mix}}(\Theta) = \sum_{k=1}^{K} \lambda_k \mathcal{L}_k(\Theta)$. The gradient update at step $t$ is governed by the Hessian of the entire mixture. Assuming the learning rate $\eta$ and training horizon $T$ are sufficiently small, we can approximate the accumulated update by expanding the gradient $\nabla \mathcal{L}_{\text{mix}}(\Theta_t)$ around $\Theta_0$:

$$\Theta^{\text{mix}} \approx \Theta_0 - \eta \sum_{t=0}^{T-1} \left( \nabla \mathcal{L}_{\text{mix}}(\Theta_0) - t\eta \nabla^2 \mathcal{L}_{\text{mix}}(\Theta_0) \nabla \mathcal{L}_{\text{mix}}(\Theta_0) \right)$$

$$= \Theta_0 - \eta T \nabla \mathcal{L}_{\text{mix}}(\Theta_0) + \frac{1}{2}(\eta T)^2 \nabla^2 \mathcal{L}_{\text{mix}}(\Theta_0) \nabla \mathcal{L}_{\text{mix}}(\Theta_0). \tag{3}$$

Substituting $\nabla \mathcal{L}_{\text{mix}} = \sum_k \lambda_k g_k$ and $\nabla^2 \mathcal{L}_{\text{mix}} = \sum_k \lambda_k H_k$:

$$\Theta^{\text{mix}} \approx \Theta_0 - \eta T \left( \sum_k \lambda_k g_k \right) + \frac{1}{2}(\eta T)^2 \left( \sum_k \lambda_k H_k \right) \left( \sum_j \lambda_j g_j \right). \tag{4}$$

Expanding the quadratic term in Eq. (4) reveals two distinct components:

$$\text{Quadratic}_{\text{mix}} = \underbrace{\sum_k \lambda_k^2 H_k g_k}_{\text{Self-Interaction}} + \underbrace{\sum_{k \neq j} \lambda_k \lambda_j H_k g_j}_{\text{Cross-Interaction}}. \tag{5}$$

**Model Merging Trajectory.** For the merging approach, we independently train $K$ experts. The update for expert $k$, trained solely on $\mathcal{L}_k$, is approximated as:

$$\Delta\Theta_k \approx -\eta T g_k + \frac{1}{2}(\eta T)^2 H_k g_k. \tag{6}$$

By merging these experts with weights $\boldsymbol{\alpha}$ set equal to the data ratios $\boldsymbol{\lambda}$:

$$\begin{aligned}
\Theta_{\text{merge}} &= \Theta_0 + \sum_k \lambda_k \Delta\Theta_k \\
&= \Theta_0 - \eta T \sum_k \lambda_k g_k + \frac{1}{2}(\eta T)^2 \sum_k \lambda_k H_k g_k.
\end{aligned} \tag{7}$$

**The Discrepancy $\Delta$.** Subtracting Eq. (7) from Eq. (4), the first-order linear terms cancel perfectly. The discrepancy is confined to the second-order terms:

$$\begin{aligned}
\boldsymbol{\Delta} &:= \Theta_T^{\text{mix}} - \Theta_{\text{merge}} \\
&= \frac{1}{2}(\eta T)^2 \left[ \left( \sum_k \lambda_k^2 H_k g_k + \sum_{k \neq j} \lambda_k \lambda_j H_k g_j \right) - \sum_k \lambda_k H_k g_k \right] \\
&= \frac{1}{2}(\eta T)^2 \left[ \underbrace{\sum_{k \neq j} \lambda_k \lambda_j H_k g_j}_{\text{Cross-domain Interference}} + \underbrace{\sum_k (\lambda_k^2 - \lambda_k) H_k g_k}_{\text{Self-domain Scaling}} \right].
\end{aligned} \tag{8}$$

Equation (8) explicitly characterizes the error introduced by the weight mixing proxy. The error consists of two parts with distinct physical interpretations:

1. **Cross-domain Interference:** The term $\sum_{k \neq j} \lambda_k \lambda_j H_k g_j$ represents the distortion of the gradient direction of domain $j$ by the curvature of domain $k$. In data mixing, the optimization trajectory of domain $j$ is dynamically modulated by the Hessian geometry of domain $k$. Model merging, by virtue of independent training, decouples these trajectories, implicitly removing the cross-term curvature effects.

2. **Self-domain Scaling:** The term $\sum (\lambda_k^2 - \lambda_k) H_k g_k$ represents a mismatch in the effective step size for each domain, reflecting curvature-induced saturation.

### B.2. Empirical Observation of Task Orthogonality and Curvature

While we rely on the first-order dominance for ranking mixtures, we provide empirical observations to further characterize the optimization landscape. To quantify the cross domain interference , we define the relative effective curvature $\gamma_{kj}$:

$$\gamma_{kj} = \frac{\|H_k g_j\|/\|g_j\|}{\|H_k g_k\|/\|g_k\|}, \tag{9}$$

where $\gamma_{kj}$ measures the normalized curvature response of domain $k$ induced by the gradient direction of domain $j$, relative to its own self-curvature. We compute the pairwise matrix $M_{k,j} = \gamma_{kj}$ for four distinct mid-training domains using Hessian-vector products (HVP).

As visualized in Figure 7a, cross-domain interference is smaller than intra-domain dynamics, resulting in a relatively small approximation error. We also visualize the cosine similarity matrix of the task vectors $\Delta\Theta_k = \Theta_k - \Theta_0$ in Figure 7b, which further illustrates the near-orthogonality of domain-specific updates in parameter space.

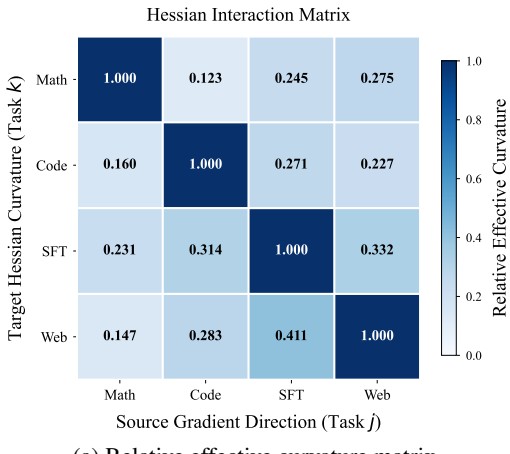

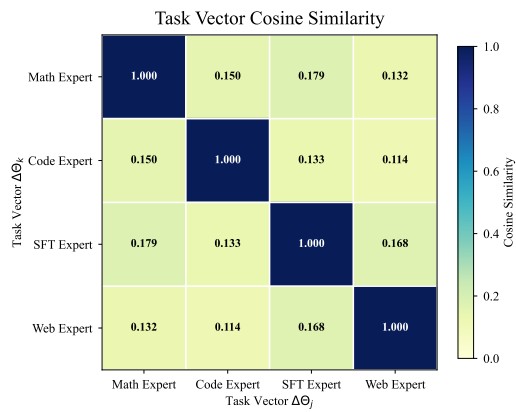

(a) Relative effective curvature matrix.

(b) Task vector cosine similarity.

*Figure 7.* (a) The matrix exhibits a diagonally dominant structure, where the self-induced curvature consistently outweighs cross-domain interference. (b) The generally low similarity scores indicate that the accumulated parameter updates tend to traverse distinct directions in the parameter space.

---

**Algorithm 1** Iterative Human-in-the-loop Optimization (Manual)

---

1: **Input:** Initial expert prior $\boldsymbol{\lambda}_{\text{prior}}$, Budget $N$ trials.
2: $\mathcal{S}_{\text{candidates}} \leftarrow \{\boldsymbol{\lambda}_{\text{prior}}\}$
3: **for** $i = 1$ **to** $N$ **do**
4:     $\Theta_i \leftarrow \text{Train}(\boldsymbol{\lambda}_{\text{current}})$
5:     $score_i \leftarrow \text{Evaluate}(\Theta_i)$
6:     Update best score and $\boldsymbol{\lambda}^*$
7:     {Human experts adjust ratio based on specific metric drops (e.g., if Math drops, increase $\lambda_{\text{math}}$)}
8:     $\boldsymbol{\lambda}_{\text{next}} \leftarrow \text{HeuristicAdjust}(\boldsymbol{\lambda}_{\text{current}}, \text{feedback})$
9:     Add $\boldsymbol{\lambda}_{\text{next}}$ to $\mathcal{S}_{\text{candidates}}$
10: **end for**
11: **Return** $\boldsymbol{\lambda}^*$

---

## C. Baselines and Computational Cost Analysis

In this section, we provide detailed specifications of the baseline methods compared in our experiments and present a quantitative analysis of their computational costs. Table 6 summarizes the resource consumption associated with each strategy.

**Manual Tuning.** This baseline represents the standard high-resource industrial practice. To ensure a strong baseline, we simulate a human-in-the-loop tuning process rather than a random search as shown in Algortithm 1. Starting from a production-verified prior distribution, we conduct an iterative refinement process involving full-scale training runs (200B tokens each on the 8B model). In each iteration, human experts analyze the benchmark feedback from the previous run and heuristically adjust the mixing ratios (e.g., increasing the math proportion if math scores stagnate). The final reported score corresponds to the best-performing mixture found throughout this resource-intensive search. In our study, we conduct 10 rounds of iterative refinement. The manual approach, while capable of yielding strong results through extensive experimentation, suffers from inherent limitations: it relies heavily on engineers' intuition and domain expertise to guide iterative decisions, introducing significant cost and uncertainty. Due to these practical constraints, it is objectively difficult to guarantee that the manually tuned mixture reaches the true optimum.

**Adapted RegMix.** The original RegMix (Liu et al., 2025) relies on training numerous small-scale proxy models (1M parameters) to fit a regressor. However, such tiny proxies are ineffective for capturing the emergent capabilities (e.g., complex reasoning) required in mid-training. To enable an effective comparison and reuse our experimental runs, we adapt RegMix to our setting by training the performance predictor using the outcomes of the 10 full-scale manual runs described above.

**CLIMB & Scaling Laws.** For CLIMB (Diao et al., 2025) and Data Scaling Laws (Shukor et al., 2025; Ye et al., 2025), due to computational resource constraints, we do not fully reproduce their pipelines. We report the experimental configurations and search budgets in their original papers in Table 6.

*Table 6.* Cost comparison between MergeMix and existing data mixing strategies. Inference overhead is negligible.

| Method | Model Size ($N$) | Train Tokens ($D$) | Runs | Equivalent Cost ($N \times D \times$ Runs) | Relative Cost |
|---|---|---|---|---|---|
| Manual | 8B | 200B | 10 | 16,000 | 100× |
| Adapted RegMix | 8B | 200B | 10 | 16,000 | 100× |
| CLIMB | 350M | 40B | 112 | 1,568 | 9.8× |
| Scaling Laws | ∼1B(sum) | 30B | 40 | 1,200 | 7.5× |
| **MergeMix** | **8B** | **5B** | **4** | **160** | **1.0×** |

## D. Implementation Details of Model Merging

In this work, model merging is employed in three distinct contexts: (1) merging domain-specific experts to search for optimal data ratios; (2) merging checkpoints during training to simulate learning rate annealing (following the WSM schedule (Tian et al., 2025)); and (3) merging the top-16 checkpoints to obtain the final model. For all these operations, we employ standard element-wise arithmetic averaging (Izmailov et al., 2018) across all model parameters, including the attention layers, MLP layers, expert weights, and the router/gate parameters, without requiring additional alignment steps. Our empirical results confirm that this simple averaging strategy is highly effective in our setting.

## E. Capacity Landscape

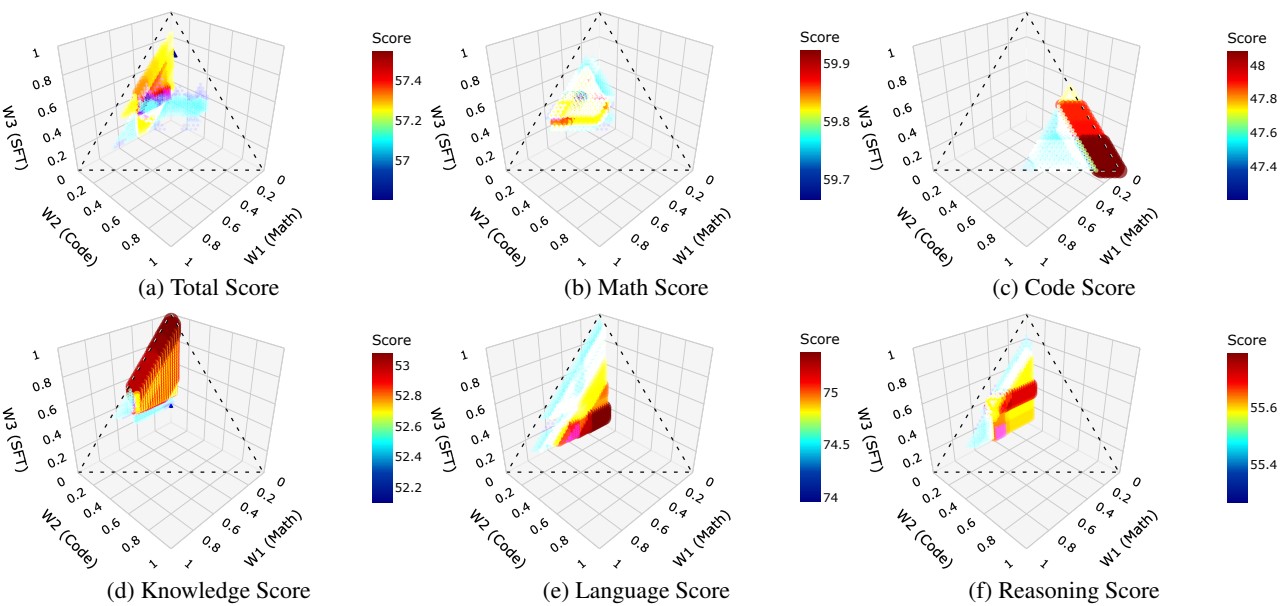

(a) Total Score  (b) Math Score  (c) Code Score

(d) Knowledge Score  (e) Language Score  (f) Reasoning Score

*Figure 8.* Visualizing the performance landscape in the weight space. Since the weights of the four domains sum to 1 ($\sum w_i = 1$), we project the 4D simplex into a 3D view. Warmer colors (red) indicate higher performance, revealing distinct topological optima for different capabilities. To enhance clarity, we visualize only the top 15% of sampled points by performance.

## F. Evaluation Details

Figure 9 reports the detailed training dynamics for each benchmark in the main experiment.

## G. Sensitivity to Expert Training Horizon

While our theoretical analysis provides a local first-order approximation under a short-horizon, shared-initialization regime, we empirically demonstrate that MergeMix remains robust over extended training horizons.

To explicitly analyze the sensitivity of MergeMix to the expert training horizon, we ablate the expert training token budget ($D_{\text{expert}}$). Figure 10 illustrates the overall Spearman rank correlation ($\rho$) between actual mixed-data training outcomes and MergeMix predictions on the 8B model, as $D_{\text{expert}}$ scales from 1B to 50B tokens.

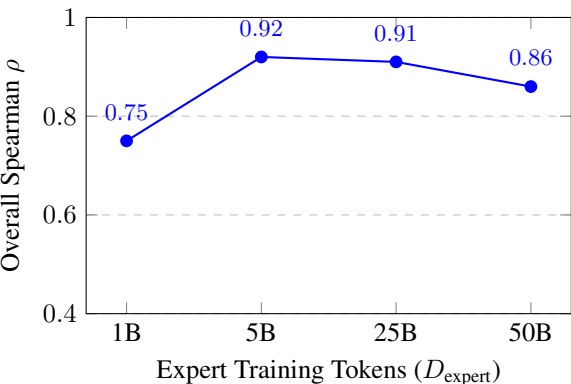

*Figure 10.* Overall Spearman rank correlation ($\rho$) between actual mixed-data training outcomes and MergeMix predictions across different expert training token budgets.

The results show that while extremely small data budgets (e.g., 1B tokens) may yield insufficient signal, the method is **highly robust across a broad range (5B–50B tokens)**. This empirically validates that the parameter-space geometry remains stable and does not easily drift into disconnected loss basins within this extended regime. Consequently, MergeMix is easy to deploy in practice and does not require extensive hyperparameter tuning for the training horizon.

## H. Data Mixing vs. Merged Models for Final Training

While Figure 3 shows that the merged model appears superior to data mixing, it is important to note that the data-mixed baseline in Figure 3 was trained with a *constant learning rate* (no annealing). In contrast, model merging inherently suppresses training noise, leading to smoother and seemingly higher performance curves under this specific setting.

To rigorously investigate whether model merging can replace data mixing for the final training stage, we provide an additional experiment comparing fully optimized pipelines under an equal token budget (600B tokens):

- **Data mixing + LR annealing:** A single model trained on 600B tokens with a standard learning rate decay.

- **Model merging:** Four domain experts trained independently (150B tokens each, totaling 600B), followed by a search over merging coefficients to find the best performance.

*Table 7.* Performance comparison between data mixing and model merging pipelines.

| Method | Avg. Performance |
| --- | --- |
| Model Merging | 67.8 |
| Data Mixing | **68.3** |

As shown in Table 7, data mixing ultimately achieves better downstream performance under practical settings. These results suggest that data mixing remains the preferred final training framework, while model merging is best positioned as a high-fidelity, low-cost proxy for efficiently identifying near-optimal mixture ratios.

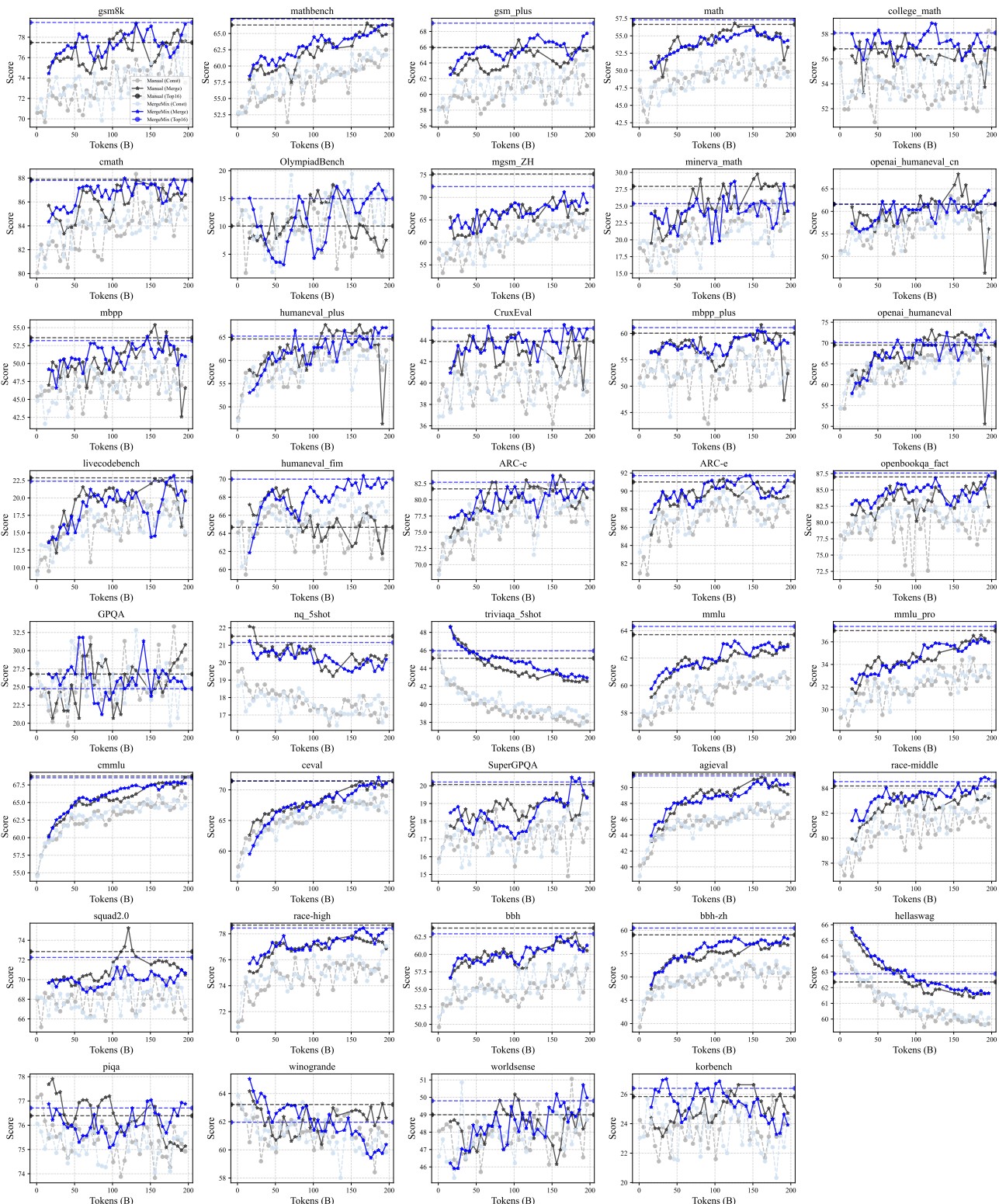

*Figure 9.* Detailed training dynamics on each individual benchmark.

