# OpenReview forum: "MergeMix: Optimizing Mid-Training Data Mixtures via Learnable Model Merging"
_ICML.cc/2026/Conference — ICML 2026 regular_

### Official Review · Reviewer_7ZXv · 2026-03-08

**Soundness:** 4
**Presentation:** 3
**Significance:** 4
**Originality:** 3
**Overall Recommendation:** 5
**Confidence:** 3

**Summary:**

This paper proposed a new framework aimed at finding the optimal data mixing ratios on LLM mid-training stage, called MergeMix. It mathematically analyzes and proves the approximate relationship between model merging and data mixing to achieve high-fidelity, low-cost performance. Extensive experiments on LLMs validate that MergeMix achieves performance comparable to or outperforms the manual method while drastically reducing search costs.

**Compliance With Llm Reviewing Policy:**

Affirmed.

**Key Questions For Authors:**

- **(Q1)** Could the authors provide some results or analysis of the performance of increased training steps of the experts?
- **(Q2)** Could the author provide some results that replace the interpolating method with some dynamic methods?

**Limitations:**

This method now apply on LLM tasks, but does not provide performance on the VLM tasks. It would be better if this method could apply on the VLM domain.

**Strengths And Weaknesses:**

### Strenghtness

- This method propose use of model merging to replace the manual-based data mixing method, which sigificate reduce the training cost.
- This method achieved significant gains through extensive experiments across multiple dimensions, including math, code, knowledge, and reasoning.

### Weaknesses

Trainingislimitedtoashorthorizon(e.g.,fewerthan5Btokens).Thiskeepseachexpertwithinalocalneighborhood
aroundtheinitialization,preservingthegeometricalign
mentrequiredforeffectiveweightmergingandpreventing
divergenceintoseparatelossbasins

- **(W1)** As the author said, the experts with limited training ensure that the geometric alignment
ment, this may cause hard to control in other tasks.
- **(W2)** The authors compare with the vanilla interpolating model merging, but lack the ablation study and analysis comparing with some dynamic model merging methods.

---

> ### Author Rebuttal · Authors · 2026-03-28
>
> We thank the reviewer for the encouraging rating (`5: Accept`) and the positive evaluation of MergeMix’s efficacy and practical value. Below, we answer the questions and limitations point by point.
>
> ---
>
> > [Q1 / W1] Performance and analysis on increased expert training steps.
>
> To address your concern regarding the expert training horizon and geometric alignment, we ablate the expert training token budget ($D_{expert}$). Specifically, we evaluate the Spearman rank correlation ($\rho$) between actual mixed-data training outcomes and MergeMix predictions using varying $D_{expert}$:
>
> | $D_{\text{expert}}$ | Overall | Math | Code | Knowledge | Language | Reasoning |
> | :--- | :---: | :---: | :---: | :---: | :---: | :---: |
> | 1B tokens | 0.75 | 0.59 | 0.83 | 0.93 | 0.95 | 0.97 |
> | 5B tokens | 0.92 | 0.84 | 0.80 | 0.98 | 0.72 | 0.89 |
> | 25B tokens | 0.91 | 0.76 | 0.83 | 0.95 | 0.95 | 0.90 |
> | 50B tokens | 0.86 | 0.74 | 0.85 | 0.96 | 0.94 | 0.98 |
>
>
> The results show that while extremely small data budgets may yield insufficient signal, the method is **highly robust across a broad range (5B–50B tokens)**. This empirically validates that the **parameter-space geometry remains stable and does not easily drift into disconnected loss basins** within this regime. Consequently, MergeMix is easy to deploy and does not require overly strict controls or extensive hyperparameter tuning for the training horizon.
>
>
> > [Q2 / W2] Comparison with dynamic model merging methods.
>
> We appreciate this insightful suggestion. In our current implementation, we utilize standard linear interpolation rather than dynamic merging methods for two key reasons:
>
> (1) **Theoretical interpretability:** Element-wise linear interpolation aligns exactly with the first-order Taylor expansion of the loss landscape, directly supporting our theoretical framework.
>
> (2) **Training-free simplicity:** It is already highly effective for our use case while incurring negligible computational overhead.
>
> Nonetheless, we agree that exploring alternative or dynamic merging operators to find optimal data mixtures is a very promising direction, and we plan to empirically explore it in our future work.
>
> > [Limitation] Application to Vision-Language Models (VLMs).
>
> Thank you for raising this important point. Modern VLM training indeed relies on complex data mixtures (e.g., text-only, image-caption pairs, and interleaved multimodal sequences). Theoretically, the weight-to-data duality underlying MergeMix should generalize to VLMs. We intend to extend MergeMix to multimodal settings in future work and believe it could offer similar efficiency gains in large-scale VLM data recipe optimization.

---

> > ### Author Rebuttal · Reviewer_7ZXv · 2026-04-01
> >
> > My concerns have been addressed, and I keep my score. Thanks to the authors' rebuttal.

---

> > > ### Author Response · Authors · 2026-04-07
> > >
> > > Reviewer 7ZXv,
> > >
> > > We would like to thank you for your support which helped improve the paper. We are thrilled to see that our answers have resonated with you.
> > >
> > > Best Regards,
> > > The Authors

---

### Official Review · Reviewer_ARdd · 2026-03-12

**Soundness:** 2
**Presentation:** 2
**Significance:** 2
**Originality:** 3
**Overall Recommendation:** 4
**Confidence:** 5

**Summary:**

The authors propose a new data mixture optimization method by mapping it to a model merging problem across a series of domain experts. The method is backed by the optimization theory in first-order and kind of related to the linear mode connectivity theory.
The empirical results show that the performance rank between various model merging weights and data mixture weights are well-correlated, especially in the "Knowledge" type of tasks.

**Compliance With Llm Reviewing Policy:**

Affirmed.

**Final Justification:**

Thanks for the responses from the authors. The problems on the manual baseline and its mysterious "100x" costs still remain unsolved; for DoReMi and DoGE, "Because downstream benchmarks are non-differentiable" is not very convincing since though the correlation between loss and accuracy can be broken, they still serve as conventional data mixing baselines for pretraining which should be include. For Decay-sim, thanks for the explanation, but I don't think it is described anywhere in the paper.

Despite the weakness above, I still acknowledge the merit of the work especially suitable for continual pretraining stage, which directly optimize for accuracy instead of traditionally used perplexity. I am willing to increase the score to 4. Good luck!

**Key Questions For Authors:**

1. Can you provide the baseline results of conventional data mixture optimization methods, like DoReMi[2], DoGE [3] in the main experiments, and RegMix [1] in cross-scale transfer?

2. Can you provide some analysis to understand how faithful is the model merging method along the mid-training process? e.g. how far (i.e. num. of tokens) the doimain experts should be trained to ensure the model merging to be effective?

3. Model Architecture: The authors experimented with two MoE models (8B and 16B) with the same number of effective parameters (1.4B). Does the same conclusion holds for the dense model, or specifically for MoE? Also, does the cross-scale results still hold with various number of effective parameters in the MoE backbone?

4. Figure 5 shows the proposed MergeMix even outperform the high quality specialized data, which is impressive. Do you have any explanations or hypothesis on this? and could you expose the dataset used for Math and Code finetuning here?

5. What does the "Decay-sim" mean in all the figures?

[1] RegMix: Data Mixture as Regression for Language Model Pre-training.

[2] DoReMi: Optimizing Data Mixtures Speeds Up Language Model Pretraining.

[3] DoGE: Domain Reweighting with Generalization Estimation.

**Strengths And Weaknesses:**

## Strength

1. The authors propose the model merging as an efficient proxy of data mixture optimization (i.e. data resampling), which eliminates the costs of repeated retraining to find the best data mixture weights.

2. The empirical results demonstrate a strong rank correlation between the performance from model merging on various weights and the data mixture with the corresponding sampling weights.


## Weakness

1. the proposed method only theoretically reasonable when the domain-expert models are tuned from the same pretrained checkpoint and not diverge to much (i.e. on 5B tokens according to the paper). If those models are not trained enough to learn about the specialized domain or over-trained to be diverged into different basins, would the conclusion still holds? Also, this makes the proposed method limit to the mid-training phase.

2. The empirical improvement above RegMix is very marginal (Table 1).

3. If we have available strong domain experts, where the model merging are validate to be effective, why not directly apply the merged model but retrain a new one with the updated data mixture? If it's because the model merging assumption could collapse after a long training phase, could you provide some insights on when would the model merging still hold and when the collapse could happen?

4. The compared baselines not concrete enough. The "Manual" baseline mentioned in the paper, which is used as a primary baseline across the paper, is from an "internal flagship model", which is very ambiguous without any further information or model name.

5. The comparison to RegMix is not fair. RegMix [1] collects the results from hundreds of **proxy** runs then use the regressor to predict the best mixture weights. The proxy model is at least tens or hundreds times smaller than the base model, while the authors trains a same scale model with the same amount of tokens as the base model. The comparison of the computation costs is unfair. Also, the RegMix baseline is missing in the cross-scale transfer experiment (Figure 6), which should be Regmix's primary use case.

6. Miss the baseline results of conventional data mixture optimization methods, like DoReMi[2] and DoGE [3]. The ScalingLaw and ClimbMix are only mentioned in the computation cost comparison, but no results included in the further experimental results.

7. Model Architecture: The authors experimented with two MoE models (8B and 16B) with the same number of effective parameters (1.4B). Does the same conclusion holds for the dense model, or specifically for MoE? Also, does the cross-scale results still hold with various number of effective parameters in the MoE backbone?

[1] RegMix: Data Mixture as Regression for Language Model Pre-training.

[2] DoReMi: Optimizing Data Mixtures Speeds Up Language Model Pretraining.

[3] DoGE: Domain Reweighting with Generalization Estimation.

---

> ### Author Rebuttal · Authors · 2026-03-28
>
> We sincerely thank the reviewer for the detailed review and constructive feedback. We address the specific questions below:
>
> ---
>
> > [W1/Q2] Expert Training Horizon & Divergence
>
> Regarding expert training duration and potential divergence, we have provided additional experiments and analysis in our response to **Reviewer vwHd [Q1/W1]**.
>
> Furthermore, our method is not strictly limited to mid-training; for from-scratch pre-training, one only needs to train on general data for a warmup period to establish a shared initialization, after which MergeMix can be applied.
>
> > [W2] Marginal Empirical Improvement
>
> We’d like to clarify two key points:
> First, in pretraining and mid-training settings, a uniform +0.3 improvement reflects a **non-trivial advantage**. And this improvement is **consistent across all six downstream domains**, especially given that we use exactly the same high-quality datasets (only the mixture ratios differ).
> Second, and more importantly, our primary contribution lies in **dramatic cost reduction**. The adapted RegMix baseline required 10 full training runs (10 × 200B = **2T** tokens), whereas MergeMix only trained 4 domain experts (4 × 5B = **20B** tokens). This represents a 100× reduction in training compute, making optimal mixture discovery accessible without massive industrial compute budgets.
>
> > [W3] Why focus on data mixing instead of just using the optimal merged model?
>
> This is an insightful question.
> It is not due to collapse, but because actual data mixing with proper learning rate annealing still yields a higher absolute performance ceiling.
> See additional experiments and analysis in our response to **Reviewer yHge [W1]**
>
> > [W4] Ambiguity of the Manual Baseline
>
> We apologize for the ambiguity regarding the model name due to double-blind constraints. However, we would like to emphasize that the manual baseline represents a highly optimized, resource-intensive industrial standard. As detailed in Appendix C (Algorithm 1), this baseline was obtained through an extensive human-in-the-loop tuning process. Outperforming this baseline demonstrates that MergeMix can achieve a human-expert level of optimization automatically and at a fraction of the cost.
>
> > [W5/W6/Q1] Missing Baselines (DoReMi, DoGE) and RegMix in Cross-Scale
>
> 1. **DoReMi & DoGE** are designed for pre-training. They reweight data to minimize next-token prediction loss. However, lower perplexity does not reliably translate to downstream capabilities (e.g., complex coding or mathematical reasoning). Because downstream benchmarks are non-differentiable, the proxy-loss frameworks of DoReMi/DoGE cannot be directly applied here.
> 2. Regarding **RegMix**: The original RegMix relies on validation loss from tiny proxy models (e.g., 1M parameters). However, emergent capabilities like reasoning simply do not exist at this scale. To make RegMix work for mid-training capability prediction, we had to adapt it by scaling the proxy models so they could actually output meaningful benchmark scores.
>
>
> > [W7/Q3] Generalization to Dense Models and Different MoE Configurations
>
> While our experiments utilized MoE architectures due to their prevalence in modern LLM training, the core mechanism of MergeMix is fundamentally architecture-agnostic. In fact, key inspirations for our approach, such as Model Soups and Task Arithmetic, were originally discovered and extensively validated on dense architectures in prior literature. We will explicitly discuss this architectural generalization in the revised manuscript.
>
> > [Q4] Outperforming Specialized Data
>
> Thank you for this insightful question.
> We hypothesize it stems from MergeMix's capacity to capture beneficial cross-domain synergies. For instance, while pure math data directly targets mathematical reasoning, other domains, such as code (which involves logical structuring), instruction-following (which improves instruction following), can enhance math performance when appropriately weighted. MergeMix automatically discovers and leverages these complementary signals through weight interpolation, effectively constructing a richer, more balanced training signal than any single domain alone.
>
> > [Q5] Meaning of "Decay-sim"
>
> Traditional training typically requires a dedicated, computationally expensive learning rate decay phase at the end of training to achieve optimal performance. However, recent work [1,2] has shown that one can instead train with a constant learning rate throughout and approximate the benefits of LR decay by averaging the weights of recent checkpoints. This technique effectively simulates the convergence behavior of explicit annealing, yielding a final model whose performance matches that of a fully decayed run without needing a separate decay schedule. We adopt this technique simply to obtain our final fully-trained models efficiently.
>
> [1] WSM: Decay-Free Learning Rate Schedule via Checkpoint Merging for LLM Pre-training
>
> [2] Model Merging in Pre-training of Large Language Models

---

> > ### Author Rebuttal · Reviewer_ARdd · 2026-04-06
> >
> > Thanks for the responses from the authors. The problems on the manual baseline and its mysterious "100x" costs still remain unsolved; for DoReMi and DoGE, "Because downstream benchmarks are non-differentiable" is not very convincing since though the correlation between loss and accuracy can be broken, they still serve as conventional data mixing baselines for pretraining which should be include. For Decay-sim, thanks for the explanation, but I don't think it is described anywhere in the paper.
> >
> > Despite the weakness above, I still acknowledge the merit of the work especially suitable for continual pretraining stage, which directly optimize for accuracy instead of traditionally used perplexity. I am willing to increase the score to 4. Good luck!

---

> > > ### Author Response · Authors · 2026-04-07
> > >
> > > Dear Reviewer ARdd,
> > >
> > > We would like to sincerely thank you for your support. We value your suggestion and will add this discussion in the next version of the paper in addition to your suggested changes.
> > >
> > > All the best,
> > > The Authors

---

### Official Review · Reviewer_vwHd · 2026-03-13

**Soundness:** 3
**Presentation:** 3
**Significance:** 3
**Originality:** 3
**Overall Recommendation:** 4
**Confidence:** 4

**Summary:**

This paper studies the problem of optimizing data mixtures for LLM mid-training. To reduce the high cost of repeatedly training candidate mixtures, the authors propose MergeMix, a method that approximates mixture search through model merging. The approach trains a small set of domain experts from a shared initialization, linearly merges these experts with different weights, evaluates the merged models on downstream benchmarks, and fits a lightweight predictor to estimate the performance landscape and identify promising mixture ratios. The selected ratios are then used for large-scale training. The paper provides a first-order analysis to motivate why expert merging can approximate mixed-data training in a local regime and validates the method through controlled two-domain experiments as well as large-scale mid-training experiments on 8B models with transfer to 16B. Empirical results show that the proposed method slightly improves over manual and adapted RegMix baselines while substantially reducing mixture-search cost.

**Compliance With Llm Reviewing Policy:**

Affirmed.

**Final Justification:**

As I have responded, the authors have addressed most of my concerns. The rest concerns lies in W4. I maintain my original positive score, i.e., 4.

**Key Questions For Authors:**

- How sensitive is MergeMix to the choice of expert training horizon? A more explicit analysis of when the proxy stops correlating well with full training would be helpful.
- Did the authors try alternative merging operators beyond simple linear interpolation, and if so, how sensitive is the overall method to this design choice?

**Limitations:**

The paper acknowledges some assumptions behind the method, but it would benefit from a clearer discussion that the theoretical justification is local, that the method is evaluated mainly in a specific mid-training regime, and that the strongest empirical evidence is not easily reproducible by the community.

**Strengths And Weaknesses:**

### Strengths
- The paper addresses an important and practical problem in LLM mid-training, and the proposed reframing of data mixture search as expert merging plus downstream evaluation is simple, intuitive, and easy to follow.
- The method is supported by a reasonable first-order analysis that provides useful intuition for why merged experts can approximate mixed-data training in the local regime considered in the paper.
- The empirical study is fairly extensive, covering controlled validation, large-scale 8B experiments, transfer to 16B, and additional analyses of dynamic and hierarchical mixtures.
- The method appears practically useful, achieving slightly better average performance than the manual and adapted RegMix baselines while substantially reducing search cost.
### Weakness
- The theoretical support is limited to a local first-order approximation under fairly restrictive conditions, so it does not fully justify the method beyond the short-horizon shared-initialization regime studied here.
- The baseline comparison is not fully convincing against the strongest prior automated mixture optimization methods. Most comparisons are against a manual baseline and an adapted internal baseline.
- In the large-scale setup, mixture optimization is somewhat entangled with other recipe choices such as constant-LR training and checkpoint merging, which makes attribution less clean.
- The strongest results rely on private industrial-scale data and training pipelines, which limits reproducibility.

---

> ### Author Rebuttal · Authors · 2026-03-28
>
> We thank the reviewer for recognizing the practical value, soundness, and extensiveness of our study.
> We address the specific questions below:
>
> > [Q1 / W1] Sensitivity to Expert Training Horizon & Restrictive Conditions.
>
> To explicitly analyze the sensitivity of MergeMix to the expert training horizon, we ablate the expert training token budget ($D_{expert}$). Specifically, we evaluate the Spearman rank correlation ($\rho$) between actual mixed-data training outcomes and MergeMix predictions using $D_{expert}$ of 1B, 10B, and 20B tokens on the 8B model:
>
> | $D_{\text{expert}}$ | Overall | Math | Code | Knowledge | Language | Reasoning |
> | :--- | :---: | :---: | :---: | :---: | :---: | :---: |
> | 1B tokens | 0.75 | 0.59 | 0.83 | 0.93 | 0.95 | 0.97 |
> | 5B tokens | 0.92 | 0.84 | 0.80 | 0.98 | 0.72 | 0.89 |
> | 25B tokens | 0.91 | 0.76 | 0.83 | 0.95 | 0.95 | 0.90 |
> | 50B tokens | 0.86 | 0.74 | 0.85 | 0.96 | 0.94 | 0.98 |
>
>
> The results show that while extremely small data budgets may yield insufficient signal, the method is **highly robust across a broad range (5B–50B tokens)**. This empirically validates that the **parameter-space geometry remains stable and does not easily drift into disconnected loss basins** within this regime. Consequently, MergeMix is easy to deploy and does not require extensive hyperparameter tuning for the training horizon.
>
>
> > [Q2] Alternative Merging Operators
>
> We appreciate this insightful question. In our current implementation, we use standard linear interpolation rather than more complex or dynamic merging operators for two key reasons:
>
> (1) **Theoretical interpretability:** Element-wise linear interpolation aligns exactly with the first-order Taylor expansion of the loss landscape, directly supporting our theoretical framework.
>
> (2) **Training-free simplicity:** It is already highly effective for our use case while incurring negligible computational overhead.
>
> Nonetheless, we agree that exploring alternative or dynamic merging operators to find optimal data mixtures is a very promising direction, and we plan to empirically explore it in our future work.
>
> > [W2] Baseline Comparisons
>
> Existing automated data mixing methods primarily target pretraining and validation loss. Transferring them to mid-training, where the focus is on downstream performance, requires substantially larger training data volumes and model scales, which would incur enormous computational costs. For example, fitting hundreds of training points is simply unaffordable. Therefore, we reused our validated effective manual method, as well as an adapted RegMix based on those results, as our baselines. These baselines are highly competitive, as they are the product of extensive ablation studies conducted by human experts. We will, however, strive to adapt and include more automated baselines for comparison in future iterations of this work.
>
> > [W3] Entanglement with Training Recipes
>
> We would like to clarify that the constant learning rate and checkpoint merging are used strictly as an evaluation protocol to simulate the final learning rate decay phase without incurring its high computational cost. Crucially, this identical evaluation protocol is applied consistently across all compared baselines. Therefore, the evaluation protocol is fully decoupled from the mixture search, and the relative performance gains are entirely attributable to the optimal data mixtures discovered by MergeMix.
>
> > [W4] Reproducibility
>
> While our experiments utilize industrial pipeline to demonstrate effectiveness at scale, the MergeMix framework is fundamentally model- and data-agnostic, making it readily adoptable by the open-source community. We are also happy to replicate and release results on open-source models and data in the future.

---

> > ### Author Rebuttal · Reviewer_vwHd · 2026-04-03
> >
> > The authors have addressed most of my concerns. The rest concerns lies in W4. I maintain my original positive score.

---

> > > ### Author Response · Authors · 2026-04-07
> > >
> > > Dear Reviewer vwHd,
> > >
> > > Thank you again for your review, and we are glad that our clarifications addressed most of your concerns. We value your suggestion and will add this discussion in the next version of the paper in addition to your suggested changes.
> > >
> > > All the best,
> > >
> > > The Authors

---

### Official Review · Reviewer_yHge · 2026-03-17

**Soundness:** 2
**Presentation:** 3
**Significance:** 4
**Originality:** 2
**Overall Recommendation:** 4
**Confidence:** 5

**Summary:**

The paper focuses on how to efficiently determine a data mix, the proportions in which different training dataset should be sampled, during LLM mid-training. Existing approaches train many proxy models on different mixes, which can be expensive. Instead, the proposed method, MergeMix, only trains one model per domain. Then, they construct and evaluate models created by merging these domain-specific models in different ratios. A regression model is fitted on these merged models to predict the performance of a model merged with any candidate ratios. Then, the final proposed data mixture is computed by optimizing predicted performance. Empirical results on 8B and 16B MOEs show that MergeMix achieves strong performance with lower costs than manual tuning.

**Compliance With Llm Reviewing Policy:**

Affirmed.

**Key Questions For Authors:**

1. Can you provide ablations that do not use the regression model and instead search over the merged models to propose a final mixture?
2. Can you evaluate an additional RegMix baseline that operates with the exact same mixing ratios (grid search, N=40) as MergeMix, etc.? I think this will be an insightful comparison, isolating merging versus mixing.
3. What is the rank consistency between results at the 5B token and 200B scales?
4. Can you provide additional versions of Figures 2 and 3 with other pairs of domains, e.g., language?
5. For the cross-scale transfer of data mixtures in section 4.4, were both models trained on the exact same data and the same amount? Or were they both trained so that the ratio of tokens to parameters is consistent, e.g., Chinchilla optimal?

**Limitations:**

Limitations are not discussed

**Strengths And Weaknesses:**

**Strengths**

Soundness:
- The paper justifies merging with theoretical analysis, showing that the second-order terms are negligible and interactions among domains are minimal.
- Experiments are conducted to demonstrate the rank correlation between mixing and merging, supporting the validity of the method.

Significance:
- The paper focuses on a large scale study and conducts data mixing experiments in the context of actually developing a language model. For instance, the experiment on dynamic versus static mixes is useful simply because these sorts of experiments are quite sparse in the literature. The hierarchical setup is also realistic with mid-training domains divided by skill, and skills split into fine-grained sub clusters. Lastly, the manual tuning baseline where the data team iterates on various mixes is valuable for a realistic assessment of data mixing algorithms.

**Weaknesses**

Soundness:
- Given the results that merged models perform around 2 pts better than data mixing (Figure 3), I am curious why the problem is focused on data mixing at all. In particular, if one is not transferring mixes across model sizes, why not simply train (e.g., on the optimal mix) starting from the optimal merged model? (I may be misunderstanding something though, see my comment in 'Presentation'.)
- I am curious about ablations that do not use the regression model. Rather than fitting a model to predict performance of new mixes, you could conduct a search over the merged models that you use for regression. While this has higher cost (running inference on every candidate), this is still significantly cheaper than training models on different mixes.
- I am also curious about the choice of regression model, and validating it with some regression fit experiments. Merging seems to imply that domains have linear interactions so it would be interesting to evaluate regression models that use parametric mixing laws (such as Data Mixing Laws and BiMix).
- I have some concerns with the adapted Regmix baseline. It uses the 10 mixes from the manual baseline, which might all be concentrated in some locality of the mixture space and prevent learning a good regression model (10 mixes is also not that many samples to learn from for a feature vector of 4 domains). In contrast, standard RegMix samples from a Dirichlet distribution, centering on a prior while ensuring sufficient coverage of the mixture space. Therefore, I am wondering if doing RegMix over the $N=40$ grid search ratios used in MergeMix could be a valuable baseline, which directly compares the use of merged models versus mixed models while holding all else constant.
- MergeMix states that it is cheaper because $D_{expert}$ (5B tokens) is less than $D_{trial}$ (200B tokens). Are there ablations that investigate the rank consistency between these two settings?
- For 4.2 (Figures 2 and 3), it would be good to have results for merging versus mixing on other pairs of domains, e.g., language (since language has a lower rank correlation in Table 2).
- Regarding the hierarchical setup, only the top-down method is evaluated, and not the bottom-up approach. It would also be more convincing if a flat mixing approach across all of the fine-grained domains was evaluated as an oracle.


Presentation:
- It was slightly unclear what the final output is. Does MergeMix produce a set of mixture ratios, which are used to start midtraining from the original checkpoint? I don't understand how the final model merging (across the 16 checkpoints, and the learning rate annealing) come into play. I also do not fully understand what "learning rate annealing (simulated by merging the most recent 20B tokens)" means.
- For the cross-scale transfer of data mixtures in section 4.4, were both models trained on the exact same data and the same amount? Or were they both trained so that the ratio of tokens to parameters is consistent, e.g., Chinchilla optimal?
- Were the benchmarks in section 4.1 the ones that MergeMix optimizes on (e.g., the $M$ capability domains)? Did scores across benchmarks need to be normalized before being aggregated?

Originality: should have more detailed discussion of comparison to Merge to Mix (Tao et. al.) The general idea is somewhat similar, although I understand the differences.

---

> ### Author Rebuttal · Authors · 2026-03-28
>
> We thank the reviewer for the positive assessment of our work and for recognizing the soundness, significance, and practical value of our large-scale study. We address your specific questions below:
>
> ---
>
> > [W1] Data mixing vs. Merged Models?
>
> While Figure 3 shows that the merged model appears superior, it is important to note: the data-mixed baseline here was trained with a constant learning rate (no annealing), whereas model merging inherently suppresses training noise.
>
> We provide additional experiments comparing fully optimized pipelines. We compare:
> - **Data mixing + LR annealing**: A single model trained on 600B tokens with a standard learning rate decay.
> - **Model merging**: Four domain experts trained independently (150B tokens each, totaling 600B), followed by a search over merging coefficients to find the best-performance.
>
> | Method| Avg. Performance |
> |--|---|
> | Data Mixing | **68.3** |
> | Model Merging   | 67.8|
>
> These results show that data mixing remains the preferred final training framework, while model merging is positioned as a high-fidelity, low-cost proxy for efficiently identifying near-optimal mixture ratios.
>
> > [Q1/W2] Search-only over Merged Models
>
> In practice, our implementation is already essentially a hybrid of predictive regression and actual search. A pure grid search (e.g., using a step size of 0.1 across 4 domains) yields hundreds of candidate mixtures. Although inference is much cheaper than training, evaluating this many merged models across our extensive suite of downstream benchmarks still incurs substantial evaluation overhead. The regression model serves as an efficient, low-cost filter that maps the performance landscape and identifies the most promising candidates. We then empirically verify these top candidates by actually merging and evaluating them. This synergy ensures both reliability and computational feasibility.
>
> > [Q2/W4] Additional N=40 RegMix Baseline
>
> The original RegMix relies on the validation loss of tiny proxy models (e.g., 1 MB) trained on minimal data. However, mid-training targets complex downstream capabilities (e.g., math, coding). At such a small scale, it is difficult to capture meaningful performance. To properly adapt RegMix for downstream performance prediction, we must use fully trained models. Evaluating an $N=40$ grid search via full training runs (200B tokens each) would require **8T tokens** of training, which is computationally prohibitive. MergeMix's primary advantage is bypassing this cost, requiring only $N_{\text{domains}} \times D_{\text{expert}} = \mathbf{20B}$ tokens.
>
>
> > [Q3/W5] Rank Consistency between 5B and 200B Token Scales
>
> We interpret this question as asking: *How well do early-training (5B-token) checkpoint rankings predict final (200B-token) performance in actual data mixing training?* To answer this, the Spearman rank correlations per domain are:
>
> | Domain | Spearman \(\rho\) |
> |---|-|
> | Overall| 0.82 |
> | Math| 0.36 |
> | Code| 0.94 |
> | Knowledge   | 0.94 |
> | Language  | 0.65 |
> | Reasoning   | 0.69 |
>
> (Note: If you were inquiring about the effect of varying \(D_{\text{expert}}\) instead, please refer to our response to **Reviewer 7ZXv, [Q1]**).
>
> > [Q4/W6/W7] Additional Experiments with Other Pairs of Domains & Flat Mixing
>
> We sincerely appreciate this valuable suggestion. Unfortunately, due to tight compute budgets and the limited rebuttal timeframe, we are unable to finish additional pairwise domain ablations or a full flat-mixing baseline  at this stage. We fully agree that these experiments would further strengthen the analysis, and we commit to including them in the future version.
>
> > [W8] Final Output of MergeMix and "Simulated" LR Annealing
>
> To clarify, the final output of the MergeMix pipeline is a **set of optimal data mixture ratios**. These ratios are subsequently used to execute the actual mid-training run starting from the original checkpoint.
>
> Regarding simulated LR annealing, see our response to **Reviewer ARdd [Q5]**.
>
> > [Q5/W9] Cross-scale Transfer Training Details
>
> Both models were trained by sampling from the exact same datasets. Specifically, the 8B model was trained on 200B tokens, and the 16B model was trained on 300B tokens. While not strictly Chinchilla-optimal (which dictates ~20x tokens per parameter), the token budgets (25x for 8B and 18.75x for 16B) align closely with standard empirical compute constraints and modern scaling practices.
>
> > [W10] Benchmarks and Score Aggregation
>
> Yes, the benchmarks listed in Section 4.1 are the ones that MergeMix optimizes over. We directly aggregate the raw scores without normalization.
>
>
> > [W11] More Discussion on Related Work
>
> Thank you for the helpful feedback. We will add a paragraph to clarify both the conceptual similarities and key distinctions.

---

> > ### Author Rebuttal · Reviewer_yHge · 2026-04-06
> >
> > Thank you for your response, which has clarified a majority of my concerns.
> >
> > [Q2/W4] Additional N=40 RegMix Baseline
> >
> > Apologies if I was not clear on this - I don't think you necessarily need to train models from scratch; you could do an adaptation of RegMix where you midtrain N=40 models on the mixes from the grid search (i.e., instead of $N_{\text{domains}} \times D_{\text{expert}}$, do $N \times D_{\text{expert}}$). My main point was that the Regmix baseline in the paper and MergeMix learn from different data mixtures.

---

> > > ### Author Response · Authors · 2026-04-07
> > >
> > > Dear Reviewer yHge,
> > >
> > > Thank you again for your thoughtful review. We’re glad that our clarifications addressed most of your concerns.
> > > We greatly appreciate your suggestion and will do our best to incorporate additional experiments or analyses as feasible, and we will revise the paper accordingly based on your feedback.
> > >
> > > All the best,
> > >
> > > The Authors

---

### Decision · Program_Chairs · 2026-04-30

**Decision:**

Accept (regular)

**Comment:**

This paper propose "MergeMix," a framework that bypasses the need to repeatedly train models on different data mixtures. Instead, it trains domain-specific experts on a small number of tokens, merges them using linear interpolation, and uses the downstream performance of the merged models as a low-cost proxy to find the optimal data mixing ratios. The authors provide a first-order theoretical justification and empirical validation on 8B and 16B parameter MoE models, demonstrating a massive reduction in compute costs with performance comparable to or slightly better than exhaustive manual tuning. The paper received uniformly positive reviews. Following the rebuttal, particularly the new ablation showing the robustness of the expert training horizon , all reviewers acknowledged that their concerns were mostly or fully resolved. That said, one of the reviewers raised concerns regarding reproducibility in an industrial pipeline and the conceptual framework looks architecture-agnostic, and the authors have committed to releasing results on open-source models. The authors are strongly encouraged to prominently incorporate the ablation study on the expert training horizon, the expanded discussion on baselines (why DoReMi/DoGE are unsuitable for mid-training capability prediction), and the clarifications regarding the final training recipe (LR annealing vs. constant LR eval) in the subsequent version to strengthen the paper.